# Providing psychological support to people impacted by terminal illness: A mixed methods study of hospice staff perceived competence, confidence and training needs

Fatema Zavery[1]☉, Sulafa Fakhreddin[1]☉, Sarah Huibregtse Van Loon[1]‡, Mhairi McDougall[1]‡, Anne Finucane[1,2]*☉

1 Clinical Psychology, School of Health in Social Science, University of Edinburgh, Edinburgh, United Kingdom, 2 Marie Curie Hospice Edinburgh, Edinburgh, United Kingdom

☉ These authors contributed equally to this work.
‡ SHVL and MMC authors also contributed equally to this work. FZ and SF are joint first authors.
* a.finucane@ed.ac.uk

## Abstract

### Background

Psychological distress is common amongst people with a terminal illness. While mental health specialists have a role in assessment and management of those with complex psychological problems, hospice clinicians provide psychological support to most patients and families.

### Aim

To describe current practices relating to the provision of psychological support by hospice clinicians, and to explore perceived competence, confidence and training needs in relation to this aspect of their role.

### Design

We used a parallel mixed methods research design. An online questionnaire consisting of closed and free-text questions was emailed to 273 hospices in the UK and the Republic of Ireland between May and June 2023.

### Setting/Participants

Participants included nurses, doctors, and allied health professionals employed by a hospice. Quantitative data was analysed descriptively using SPSS 27, and free-text data was analysed thematically guided by the framework method.

**Data availability statement:** All relevant data are within the manuscript and its Supporting Information files.

**Funding:** The authors declare no specific funding for this research. AF is funded by a Marie Curie Senior Research Fellowship (MCRFS-20-101). www.mariecurie.org.uk.

**Competing interests:** The authors have declared that no competing interests exist.

## Results

151 hospice staff completed the questionnaire. Most (81%) reported that they regularly screen for psychological distress, but clinical judgement, as opposed to use of a validated screening tool, was most common. Respondents reported confidence and competence in many areas. Overall, 72% strongly agreed they were willing to explore difficult subject matter. However, only 25% strongly agreed they were confident in differentiating level of psychological need, and 36% reported they could not arrange appropriate psychological support when needed. Almost all (95%) agreed that training in psychological support would enhance their practice. Individual and family factors such as denial, communication challenges and family conflict were barriers to providing psychological support. Systemic factors were time constraints, prioritisation of physical symptoms and limited access to mental health specialists.

## Conclusion

Hospice staff report that they are confident in providing basic psychological support. However, there was a desire for further training in this aspect of care. Clearer guidance on referral criteria for specialist psychological support is warranted.

## Introduction

Palliative care aims to improve the quality of life of people impacted by life-threatening or terminal illness by addressing their holistic needs, encompassing physical, psychological, social, and spiritual well-being. While effective management of pain and physical symptoms is crucial, psychological and non-physical concerns significantly impact patients' overall quality of life [1]. Common psychological concerns experienced by patients with terminal illnesses range from general distress to clinically diagnosed psychological problems [1–4]. Psychological distress includes feelings of sadness, anxiety, anger, frustration, fear or worry, and is commonly experienced following the diagnosis of a life-limiting illness [5–7]. A retrospective review of referral documentation relating to patients referred to a UK hospice service found that psychological distress was documented for 59% of those being referred [4] Furthermore, more serious mental health problems are common, and 29% of patients experience clinically significant levels of depression, anxiety, adjustment disorder or low mood [2].

International guidelines emphasise the importance of psychological support for people with a terminal illness and offer guidance on its provision [8–10]. One of the most frequently cited models is that developed in the UK by the National Institute for Health and Care Excellence [8]. This is a four-level model describing recommendations for psychological assessment and support for people with cancer, including advanced cancer. It specifies the appropriate psychological skills, techniques and interventions that should be available depending on the patient's level of need. (Fig 1). According to this model, all health and social care professionals should be able to

| Level | Group | Assessment | Intervention |
|-------|-------|-----------|-------------|
| 1 | All health and social care professionals | Recognition of psychological needs | Effective information giving, compassionate communication and general psychological support |
| 2 | Health and social care professionals with additional expertise | Screening for psychological distress | Psychological techniques such as problem solving |
| 3 | Trained and accredited professionals | Assessed for pychological distress and diagnosis of some psychopathology | Couselling and specific psychological interventions such as anxiety management and solution-focused therapy, delivered according to an explicit theoretical framework |
| 4 | Mental health specialists | Diagnosis of psychopathology | Specialist psychological and psychiatric interventions such as psychotherapy, including cognitive behavioural therapy (CBT) |

**Fig 1. National Institute for Health and care Excellence (NICE) recommended model of professional psychological assessment and support.**

provide general psychological support. Similarly, clinical practice guidelines in the United States from the National Coalition for Hospice and Palliative Care [10] highlight the role of the interdisciplinary team in addressing psychological and psychiatric aspects of care for those with a serious illness. While the social worker is expected to play a key role, all clinicians providing palliative care are expected to have the skills to identify and help people manage basic psychological concerns. In Australia, guidelines recommend that a psychologist is available, at least part-time, to provide support across community-based cancer services, acute settings and palliative care units [11].

Despite the recognition of psychological care as an important domain of palliative care, research evidence on the provision of psychological support in hospices is lacking. The small number of existing studies reveal broadly similar findings. Access to psychological support is insufficient [12–15]. Surveys in the UK reveal potential gaps in practice. For instance, a 2019 survey indicated that while basic psychological support was largely met, patient access to screening, assessment and psychological interventions was less consistent and patient needs often unmet or only partly met [12]. A subsequent survey in 2020 further highlighted that a significant proportion of hospice staff did not routinely screen for psychological problems and that many nurses and allied health professionals lacked confidence in providing adequate emotional support and information on available treatments [14].

To provide high-quality palliative care, health and social care staff caring for people with a life-limiting illness need to be confident and competent in their abilities to provide psychological care. Competence refers to the skills, knowledge and abilities to perform successfully or efficiently [16]. Confidence is one's appraisal or feeling of self-assurance arising from their ability to successfully perform a task or achieve an objective, and it can change as the situation or context changes [17]. It is useful to consider both competence and confidence as this allows a more comprehensive evaluation of an individual's education and training needs [17]. Training in psychological support is linked with improvements in staff confidence in detecting and managing distress [18,19], but information on the aspects of psychological support that staff find most challenging is needed to guide the development of effective and cost-effective training programmes.

Therefore, to inform the future development of psychological support for people impacted by life-limiting illness, up-to-date evidence on the role, confidence and competence of hospice staff in providing psychological support is warranted. Consequently, we sought to investigate the type of psychological support provided by hospice staff who are not employed as mental health specialists, and to explore any further training needs they perceive they may have in relation to this aspect of their role.

## Methods

### Design

We used a parallel mixed method research design, which entails the simultaneous collection of qualitative and quantitative data [20] We used the questionnaire variant of the convergent mixed methods design, which involves using a questionnaire with both closed-ended and open-ended questions but does not include a complete qualitative data collection [21] This was a pragmatic decision, enabling the collection of both data types to understand the phenomenon of interest within one study. This design allowed for further elaboration and interpretation of quantitative data based on qualitative findings. Our study is reported in line with the Checklist for Reporting Results of Internet E-Surveys (CHERRIES) guidelines [22].

### Participants

#### Eligibility

Eligible respondents included health and social care professionals based in an independent hospice in the United Kingdom or Republic of Ireland who provide direct care to terminally ill patients. We excluded volunteer staff and mental health specialists. We did not seek to recruit from hospital-based palliative care units, as these are managed by the National Health Service and would have required additional research governance processes that could not be completed in the time available for the study.

#### Sample size

An estimated sample size of a minimum of 172 respondents was required to detect Cohen's medium effect size (d = .5) with a power of .90 at the .05 significance level. Gpower 3.1.9.7 was used to calculate the sample size. We sought to detect a difference between means of total perceived competence/confidence across subgroups, including roles and years of experience. Sample size was calculated based on an expected medium effect size, as detected by similar research [23,24].

#### Questionnaire development

We developed an online questionnaire informed by national guidelines on improving supportive and palliative care for adults with cancer [25]. The aim of the questionnaire was to investigate current practices relating to the provision of psychological support by hospice clinicians, and to explore confidence and training needs in relation to this aspect of their role. The questionnaire draft was screened and refined by a hospice-based clinical psychologist, three community palliative care clinical nurse specialists, a research nurse, and a public representative. Twenty-one questions were chosen for the final questionnaire to fit a timeframe of 10–15 minutes. The final questionnaire comprised four sections: (i) the management of psychological needs, (ii) barriers to recognising and assessing psychological support needs, (iii) referral for specialist psychological support and (iv) education and training needs. The final section consisted of questions on demographic information. Placing demographic information at the end sought to ensure that participants focused first on the main questionnaire content and were less likely to bias their responses in line with any perceived expectations linked with demographics or participant characteristics. Furthermore, by that point, participants would have already seen the full questionnaire and would likely feel assured there were no unexpected or uncomfortable questions, before providing demographic details. The questionnaire is available in S1 File.

#### Participant recruitment

Study information was sent to contacts at 273 hospices, compiled from the Hospice UK and Irish Association for Palliative Care websites. An invitation email was sent to specific hospice managers asking them to share the study information with clinical staff. The manager's details were found by checking each hospice's website. The email contained the participant

information sheet and a link to the online consent form and questionnaire. One reminder email was sent after two weeks. Additionally, the research was advertised as a flyer created by the project team on social media, which contained a link to the questionnaire. Respondents could enter a prize draw to win one of five £30 vouchers by emailing the study team on completion of the survey. This way, the respondent's name and contact details were separate from their questionnaire responses.

## Data collection

Questionnaire data was collected through an online questionnaire platform supported by the University of Edinburgh (https://onlinesurveys.jisc.ac.uk/). The questionnaire was open from May 15th to June 9th, 2023. Questionnaire responses were only accessible to the research team via an anonymous password-protected platform and respondents were reminded of their right to withdraw up until completion of the questionnaire. Questionnaire responses were stored in a secure University of Edinburgh OneDrive, only accessible to the research team. We decided in advance to remove respondents with significant proportions of missing data (i.e., over 50%). 152 hospice staff responded. Data from one respondent was excluded as over half of the questions had not been answered, and 151 responses were included in the final analysis. Some missing data was found for 12 respondents. However, as most of the questions were completed for these respondents, their data were retained for analysis.

## Ethical approval

The University of Edinburgh's Health in Social Science ethics committee granted ethical approval on May 11th, 2023 (Case ID: 22–23CLPS072). Respondents were invited to read an online participant information sheet and an online consent before progressing to the questionnaire. The participant information sheet detailed that participants could withdraw from the study if they wished up to the time of survey submission. No data that could identify a participant's information was requested.

## Quantitative data analysis

Quantitative data were analysed using SPSS Version 27 and interpreted to identify any significant trends emerging from the participant responses. Descriptive statistics were used to identify trends in central tendency and dispersion. Furthermore, inferential statistical tests, including t-test and ANOVA, was used to examine trends in differences between groups. Visual representations, including bar charts were employed to illustrate any identified quantitative trends.

Measures of self-perceived competence and confidence were developed post-hoc during the analysis phase. The details are below:

*Post-hoc self-perceived competence score* A score of self-perceived competence in providing psychological support was developed post-hoc from eight questionnaire items by the research team. This reflected the respondent's self-evaluation of their level of skills in psychological assessment, management, and referral. Scores could range from 8 (self-perception of low competence) to 32 (self-perception of high competence). See S2 File for full details.

*Post-hoc self-confidence score* A score reflecting self-confidence in providing psychological support was developed post-hoc from seven questionnaire items by the research team. This reflected the respondents' beliefs in their capability to perform a task. Scores could range from 7 (low confidence) to 28 (high confidence). See S2 File for full details.

An independent samples t-test was conducted to investigate differences in mean perceived competence and self-confidence scores between individuals who had and had not received training in providing psychological support. The independent samples t-test was chosen as it is the most appropriate test to compare the means of two independent groups. A between groups One-way analysis of equal variance (ANOVA) was conducted to examine differences in total perceived competence/confidence scores by role (nurses; doctor/ consultants; allied health professionals or other) and years of experience (>11 years, 6–10 years, 1–5 years, < 1 year). Assumptions of equal variance were tested using a

Leven's test. Inspections of Kolmogorov-Smirnov tests revealed that normal distribution assumptions were not always met (p-values were significant). However, the sample size for most variables was sufficiently large (>30) based on central limit theorem guidance [26]. One-way ANOVA was also used as this is the most appropriate test when comparing three or more groups with multiple categories while controlling for a Type 1 error.

### Qualitative data analysis

Data from open-ended questions were imported into MS EXCEL and analysed thematically informed by the framework method [27,28]. All members of the research team were involved. The process followed the following steps: (i) familiarisation with the data through reading and re-reading; (ii) initial coding independently by four members of the team, (iii) development of initial themes through full team discussion of codes identified, (iv) merging and organising themes through full teams discussion in light of our research questions. By involving all team members and comparing perspectives on the data generated, it was possible to minimise the likelihood of bias associated with one individual's interpretations.

### Data synthesis

We structured our qualitative and quantitative findings around four pre-determined areas: assessment, management, referral, and education and training needs. The presentation of both qualitative and quantitative findings allowed a richer understanding of hospice staff perspectives on how psychological care is provided in practice, with examples of barriers and enablers in relation to providing this care.

## Results

### Sample characteristics

The mean age of respondents was 47 years (SD = 11.4; range 22–72), and 88% were female. Most had over 5 years' experience working in palliative care in a hospice setting, and over half had over 10 years' experience (Table 1).

### Perceived prevalence of psychological distress amongst patients receiving palliative care

Most respondents (84%) reported that more than half of all patients they see present with psychological distress. Psychological concerns commonly encountered by staff were anxiety (97%), adjustment disorder (91%), depression (85%), and relationship difficulties (60%). Other common concerns were noted by 26% of respondents in an open text box. These included fear of the future, spiritual distress, loss of control, worry about leaving family, and disrupted family dynamics.

### Assessment of psychological support needs

Most respondents (81%) reported regularly assessing patients for psychological distress. However only 15% reported that they assess patients' psychological support needs on first assessment. Most assessments were based on clinical judgement (58%), with just under half of respondents using a formal assessment tool (42%). Among those using assessment tools, the most used were the Integrated Palliative Care Outcome Scale (IPOS) (42%) and Hospital Anxiety and Depression Scale (HADS) (32%).

Qualitative data revealed barriers to psychological needs assessment: time constraints, denial of diagnosis, communication challenges and the presence of others. Time constraints and clinical pressure meant that staff often needed to make quick judgements on psychological support needs:

> *"With significant time constraints and pressures, some colleagues are quick to apply a psychological 'label', rather than sit and talk with the patient"* (Respondent 132, Chaplain)

**Table 1. Sample Characteristics.**

| Characteristics | Percentage |
|---|---|
| Gender identity (n = 151) | |
| Female (n = 133) | 88.1% |
| Male (n = 16) | 10.6% |
| Prefer not to say (n = 2) | 1.3% |
| Region (n = 151) | |
| England (n = 98) | 64.9% |
| Scotland (n = 21) | 13.9% |
| Northern Ireland (n = 16) | 10.6% |
| Wales (n = 12) | 7.9% |
| Republic of Ireland (n = 4) | 2.6% |
| Race/ethnicity (n = 143) | |
| British (n = 118) | 82.5% |
| Indian (n = 1) | 0.7% |
| Cornish (n = 1) | 0.7% |
| Scottish (n = 9) | 6.3% |
| Mixed (n = 2) | 1.4% |
| Irish (n = 8) | 5.6% |
| European (n = 1) | 0.7% |
| Chinese (n = 1) | 0.7% |
| Welsh (n = 2) | 1.4% |
| Work setting (n = 151) | |
| Hospice inpatient unit (n = 59) | 39.1% |
| Community palliative care (n = 49) | 32.5% |
| Across settings (n = 21) | 13.9% |
| Outpatient/day therapies (n = 9) | 6% |
| Hospital palliative care inpatient unit (n = 5) | 3.3% |
| Other (n = 8) | 5.3% |
| Role (n = 151) | |
| Nurse (n = 66) | 43.7% |
| Doctor/consultant (n = 31) | 20.5% |
| Allied Health Professional (AHP)/chaplain/social work (n = 54) | 35.8% |
| Years of Experience (n = 150) | |
| > 11 years (n = 79) | 52.3% |
| 6–10 years (n = 27) | 17.9% |
| 1–5 years (n = 35) | 23.2% |
| < 1 year (n = 9) | 6% |

*Note:* Not all respondents answered every demographic question so sample sizes are slightly lower for some variables.

Denial of a terminal diagnosis and communication barriers made it difficult for staff to assess psychological distress.

"*Denial makes assessment difficult.*" (Respondent 41, Clinical Nurse Specialist)

"*...Level of understanding of patients, language barriers*" [makes it challenging for staff to conduct assessments and provide psychological support] (Respondent 63, Nurse)

The presence of others could also make the assessment of psychological support needs difficult, especially where patients try to mask their distress from those close to them, or where caregivers mask their concerns from the patient.

*"If family members are present the patient will not always open up"* (Respondent 140, Nurse)

*"where a loved one holds back their emotions as not to distress the person who is ill."* (Respondent 8, Healthcare assistant)

## Management of psychological support needs

The mean *perceived competence* in providing psychological support was 25.2 (SD = 5.03). The vast majority of respondents indicated that they were competent in the key activities relating to the provision of psychological care (See S2 File). Self-reported competence was highest in relation to discussing sensitive topics and openly discussing distressing issues and ways to manage this. Self-reported competence was lowest in relation to being able to arrange psychological support and feeling sufficiently trained to deliver effective psychological support. Less than one-third of respondents strongly agreed that they could arrange appropriate psychological support, and only 16% strongly agreed that they were sufficiently agreed to deliver psychological support (Fig 2).

A one-way ANOVA was conducted to identify the effect of role and years of experience on respondents' competence. There was no significant effect of years of experience on competence score ($F(3,143) = 1.28$, $p = .283$). There was no difference in competence score based on whether the members worked in inpatient or community roles ($F(2,144) = 0.46$, $p = .632$). A t-test was performed to compare perceived competence across individuals who received training in psychological support and those who did not. Respondents who received any training (M = 26.6) had statistically higher perceived competence scores than those who did not (M = 21.7) ($t(145) = 7.04$, $p < 0.01$).

The mean *perceived confidence* in providing psychological support was 22.8 (SD = 3.14). Most respondents 'strongly agreed' or 'somewhat agreed' they were confident in providing key aspects of psychological care (Fig 3). Respondents reported the highest levels of confidence in exploring difficult issues and initiating distressing conversations. Lower levels

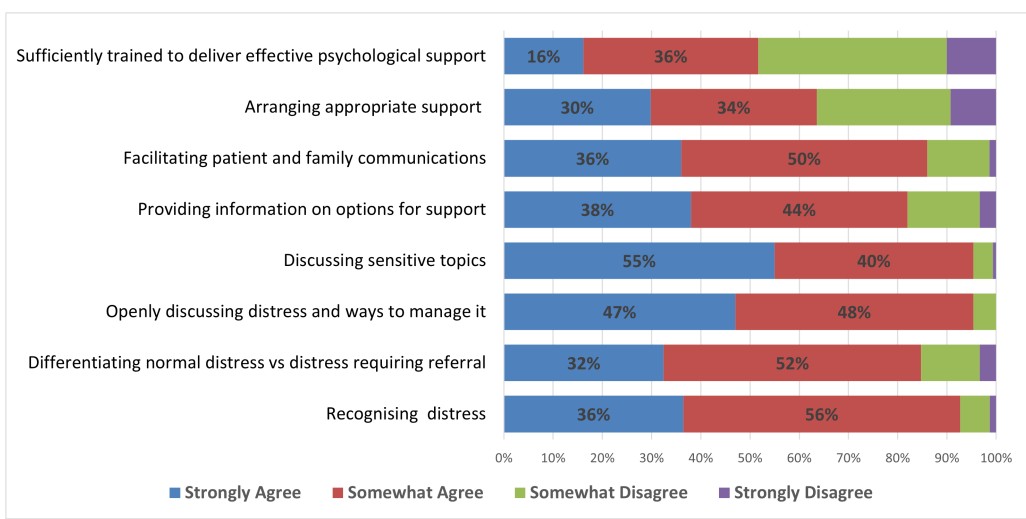

**Fig 2. Self-reported staff competence levels in distinct aspects of psychological care (N = 151).**

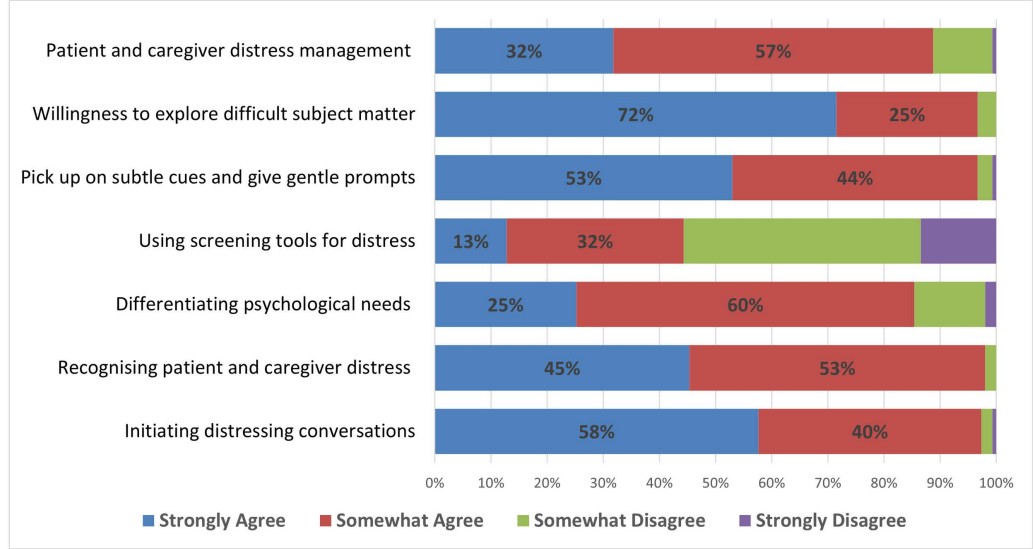

**Fig 3. Confidence of palliative care staff across various domains of psychological care (N = 151).**

of confidence related to using formal tools to screen for psychological concerns, with less than half agreeing that they were confident in this. See S2 File .

A one-way ANOVA was conducted to identify the effect of role and years of experience on respondents' confidence. Total confidence scores differed significantly across years of experience ($F(3,144) = 4.43$, $p = .005$). Bonferroni post-hoc test indicated that staff with more than 11 years of experience had higher confidence scores (M = 23.45, SD = 2.75) than staff with 1–5 years' experience (M = 21.59, SD = 3.77), 95% CI [.20, 3.52], $p = .019$. There was no statistically significant difference in total confidence score across roles, $F(2,145) = 2.27$, $p = .107$. A t-test was performed to compare perceived confidence across individuals who received training and those who did not. Respondents who received any training (M = 23.7, SD = 2.80) had significantly higher confidence scores than those who did not (M = 20.7, SD = 2.94), 95% CI [1.93, 3.70], $t(146)=5.732$ $p < 0.01$.

Respondents described barriers that could make it more difficult to provide psychological support. These included: (i) the need to prioritise the management of physical symptoms; (ii) limited time; (iii) communication challenges; (iv) environment constraints; and (v) family dynamics. Respondents reported that patients often have significant physical symptoms associated with their illness, which need to be prioritised over any coinciding psychological concerns.

*"When a patient has physical pain but there is a psychological component- sometimes it feels as though we cannot get further with the psych distress until pain is better controlled, even though addressing the former will help the latter…"* (Respondent 39, Consultant)

There was a perception that effective psychological support requires more time than is available.

*"Time is short... to be able to utilize nonpharmacological interventions effectively"* (Respondent 96, Nurse)

Communication challenges due to cognitive difficulties, language barriers, or reluctance to discuss their situation made it more difficult to provide psychological support.

*"Communication ability of patients, i.e., if they have cognitive deficits or difficulty articulating makes it difficult to manage psychological distress."* (Respondent 32, Counsellor)

*"Reluctance of patient to discuss their fears, accept their diagnosis/prognosis. Level of understanding of patients, language barriers."* (Respondent 63, Nurse)

A lack of privacy in inpatient settings can also hinder opportunities for the management of psychological concerns.

*"It's really challenging to have sensitive and tender conversation in shared bays where cloth curtains provide visual privacy but by no means any audible privacy"* (Respondent 87, Consultant)

Disagreements between patients and families about goals of care or how care is managed could also hinder access to psychological support.

*"Difficulties arise…when the patient and family are not aligned in their thinking of the patient's treatment. In some cases (families don't) consider it appropriate to consider counselling as an option"* (Respondent 138, Doctor)

In terms of facilitators of psychological support, many respondents reported that a therapeutic relationship with the patient and caregivers is the foundation of psychological care, and it is built based on trust, honesty and rapport.

*"Building a trusting rapport allows [patients and families] to express their feelings, concerns and anxieties"* (Respondent 17, Bereavement counsellor)

### Referral of patients and caregivers for psychological support

More than half of respondents (54%) reported that guidelines for psychological support referrals were unclear within their organisations. One-third of respondents disagreed with the statement that they could arrange psychological support for patients when they needed to (36%). Internally, respondents could most commonly refer patients to bereavement counsellors, complementary therapists, counsellors and the social work team for psychological support (Table 2). However, less than one-quarter of respondents (24%) could refer to an in-house clinical psychologist. Referring externally for psychological support was less common, with less than half of the respondents referring to any external services for psychological support. Where respondents did refer externally, referral to social work (48%) and clinical psychologists (43%) were most common (Table 2).

Qualitative data revealed referral challenges due to limited access to mental health specialists and a lack of clarity in relation to referral processes. Limited access to mental health specialists could be due to part-time availability:

*"Only one part-time psychologist in the palliative care team"* (Respondent 52, Consultant)

Some respondents noted that patients were more likely to be referred for psychological support if they were cared for in a hospital setting, or if they had a cancer diagnosis.

*"Having access to a regional psychologist for pall[iative] and EOLC [End Of Life Care] that could be used by all Pall[iative] and EOLC services - there is often access if the patient is in an acute hosp[ital] setting but not if the patient is in a hospice."* (Respondent 21, Occupational therapist)

*"..and very patchy support for those who don't have a cancer diagnosis"* (Respondent 52, Consultant)

Referral challenges were more often noted in relation to external referrals to specialist services:

**Table 2. Internal and external referral for specialist psychological support.**

| Internal referrals for psychological support[a] (i.e., ability to refer to professionals within the hospice for psychological support) | Percentage |
|---|---|
| Bereavement Counsellor (n=94) | 62% |
| Complementary Therapist (n=92) | 61% |
| Counsellor (n=89) | 59% |
| Social Work (n=79) | 52% |
| Day therapies (n=50) | 33% |
| Doctor/nurse trained in psychological support (n=44) | 29% |
| Clinical Psychologist (n=36) | 24% |
| Allied Health Professionals (AHP) (n=36) | 24% |
| Chaplain (n=11) | 7% |
| Senior nurse manager (n=4) | 3% |
| Psychiatrist (n=2) | 1% |
| Other (Occupational therapist/Physiotherapist, family support worker, Senior nurse managers) (n=6) | 4% |
| **External referrals for psychological support[a]** (i.e., ability to refer to professionals external to the hospice for psychological support) | **Percentage** |
| Social Work (n=72) | 48% |
| Clinical Psychologist (n=65) | 43% |
| Psychiatrist (n=44) | 29% |
| Doctor/nurse trained in psychological support (n=31) | 21% |
| Allied Health Professionals (AHP) (n=29) | 19% |
| Counsellor (n=27) | 18% |
| Bereavement counsellor (n=26) | 17% |
| Complementary therapist (n=14) | 9% |
| Psychological support provided by day therapies (n=9) | 6% |
| General Practitioner (n=7) | 5% |
| Other (n=4) | 3% |
| Chaplain/ Pastoral care (n=2) | 1% |

[a]Note: Not all respondents answered this question, and respondents could choose more than one option.

> *"Referral procedures within own workplace are effective, but referral to external agencies can be difficult, especially mental health teams."* (Respondent 10, Nurse)

Respondents also reported a lack of clarity around referral procedures and indicated a desire for further clarity on referral guidelines. There was also a perception that the waiting list for support could impact the likelihood of a referral being made:

> *"We do not have clear referral processes or criteria and waits are perceived to be long, and I think this affects how often we might refer."* (Respondent 39, Consultant)

> *"Clear guidelines on when and whom to refer to, Standard referral documentation* [would improve confidence] (Respondent 114, Nurse)

### Education and training in providing psychological support

Overall, 70% of respondents reported having received training, either formal or informal, in providing psychological support to patients and families. Of these, the majority had received informal on-the-job training (82%) (Table 3). Among those who received training in formal psychological approaches (36%), training in mindfulness approaches, counselling and Cognitive Behavioural Therapy were most often reported.

Most staff reported that training would enhance their provision of psychological support (95%). Less than half of the respondents reported their training was sufficient in psychological needs assessment (44%), management of patient psychological needs (45%), and management of caregivers' psychological needs (44%).

Qualitative findings indicated a desire for further training and supervision in providing psychological care, in particular in relation to supporting specific populations such as children and families.

"*Supervision following training in use of techniques to develop skills in delivering therapies such as CBT*" [would help]. *(*Respondent 112, Consultant)

"*Face to face training on giving psychological support to children and families within the remit of our role.*" [to improve participant confidence in the provision of psychological support]. (Respondent 145, Nurse)

## Discussion

This study describes hospice staff perspectives and their self-reported competence and confidence in providing key aspects of psychological support to patients and caregivers. Our findings indicate that hospice staff routinely assess for psychological concerns using their clinical judgment. Formal screening tools are not commonly used. Nearly all hospice staff agree that they are competent and confident in providing key elements of psychological support within the boundaries of their roles. Most staff strongly agree that they are confident in initiating and holding conversations around distressing topics. Lower levels of confidence and competence were found in relation to using formal screening tools and arranging and delivering appropriate psychological support. Time constraints were perceived as a barrier to psychological care, especially when physical symptoms need to be addressed. Despite good levels of self-reported confidence and competence in relation to discussing distressing issues, there was a consensus that training was insufficient, and further training would enhance practice.

Our findings reveal that most hospice staff routinely screen for psychological concerns based on clinical judgment. These findings support and extend findings from a national survey of palliative medicine physicians in the UK, which reported that 99% used the clinical interview to diagnose anxiety, but most did not use formal tools [29]. Our findings also

**Table 3. Type of training received.**

| Training Received | Percentage of respondents who received any training (n=106) | Percentage of all respondents (n=151) |
|---|---|---|
| Informal on the job training (n=87) | 82% | 58% |
| In-house training as part of a general training day on a range of topics (n=60) | 55% | 40% |
| Training in any formal psychological therapy (n=55) | 52% | 36% |
| In-house training focused on psychological support (n=47) | 43% | 31% |
| External training day or workshop focused on psychological support (n=44) | 40% | 29% |
| A short, unaccredited course over a week or similar (n=17) | 16% | 11% |
| Comprehensive, longer-term training, such as a certificate or diploma in psychological support/ interventions (n=14) | 13% | 9% |
| Other (n=11) | 10% | 7% |

align with evidence from inpatient palliative care settings in the United States, which found that common ways to screen for psychological distress were talking to family members (92%) and observing the patients' mood (90%); with only one-third using a formal screening tool for psychological distress [30]. Our finding contrasts with a 2020 UK survey of palliative care staff in hospice care, which reported that at least one-quarter of hospice staff did not screen or assess patients for psychological problems [14]. The lower percentage in that study may be due to the way their question was framed (i.e., in the context of referring to referral criteria) and the fact that responses were from staff in a range of roles, some of whom may not be involved in psychological needs assessment as part of their role. Our findings indicate that most clinical staff screen for psychological concerns, though this is not usually part of the patient's first assessment, in part due to time constraints and prioritisation of physical symptom management.

We found that most hospice staff report high levels of confidence and competence in relation to initiating and holding conversations on distressing issues. This suggests that many hospice-employed staff have enhanced communication skills and reflects the results from a survey of 181 hospice nurses based in the United States, which concluded that hospice nurses perceive themselves as proficient communicators with effective or very effective communication skills (85.6%) [31]. That study also indicated that 'denial' is a challenge to communication, a finding we also identified in our qualitative data, which was another barrier to psychological assessment. While hospice staff feel confident in relation to discussing difficult topics with patients and families, further communication skills training focusing on how to approach denial in a palliative care context may be valuable.

Hospice-employed staff who received any psychological training felt more competent and confident in providing psychological support than those without training. These findings align with prior research linking psychological training with objective competence [32,33]. and perceived competence [34] and allude to the value of psychological support skills training in palliative care contexts. Key targets for training were psychological needs assessment and the delivery of psychological support. There is evidence that nurse-led psychological interventions can be effective in enhancing the spiritual and psychological wellbeing of people receiving palliative care in hospices [32,35–38]. Life review, metacognitive therapy, Acceptance and Commitment Therapy, dignity therapy and Cognitive Behavioural Therapy have formed the basis for nurse-led interventions that show preliminary evidence of effectiveness in helping people manage psychological concerns and improve mental wellbeing [32,35,36,38]. Training can be brief, but as noted in our findings and the wider literature, having access to supervision after initial training is essential so that skills are developed and sustained [32,34].

Our findings indicated a lack of clear patient referral guidelines for psychological support within hospice organisations. Similar findings were previously reported; [12] the only difference being that spiritual advisors featured more strongly in the earlier study, with 78% of hospices reporting referring to a spiritual advisor for psychological support compared with only 8% reporting this in our study. The reason for the higher reported levels of access to a spiritual care provider in the earlier study is unclear, though it may partly be due to the use of the wider term "spiritual advisor" rather than "Chaplain", which may have more faith-based connotations.

Less than one-quarter (24%) of respondents could refer to an in-house clinical psychologist, while 43% could refer externally to a clinical psychologist. Similar findings were reported in a pre-pandemic United Kingdom national survey of hospice psychological support services.[12] In the latter, 19% of respondents had access to an in-house psychologist, and 41% could refer externally. While these data do not take into account the actual working hours of clinical psychologists in hospice settings, they reveal that in the United Kingdom, direct access to a clinical psychologist is available in a minority of hospices. Limited integration of clinical psychologists in palliative care settings is evidenced in many countries, including Canada, North America and Europe [11,39].

## Implications for practice

Hospice-employed staff based in the UK report that they are competent and confident in providing basic levels of psychological support in line with Level 1 national guidance for the United Kingdom (Fig.1) [8]. This support is adequate for

individuals who are not significantly impacted by anxiety, depression and adjustment problems resulting from a terminal diagnosis. However, access to more tailored psychological support for patients and families with more significant problems is limited due to little formal training in psychological support, and the lack of specialist mental health resources. One way to increase access to evidence-based psychological support for patients and caregivers is to develop and implement psychological support skills training programmes for staff, with a view to increase in-house capacity, and ensure that the clinical psychologist resource is protected for those who would most benefit. A lack of clarity around referral guidelines was identified as a problem here and elsewhere, thus, a focus on clarifying local and national guidance is warranted. The NICE guidance is now over two decades old and needs to be revisited with a view to focusing on the needs of the person rather than what the service can provide. Updated guidance that takes account of digital interventions, both self-directed and facilitated, is also needed to ensure that a wide range of support is available in a broad range of formats to meet the needs of a geographically distributed ageing population.

## Future research questions

Future research is needed to explore the perspectives of patients and caregivers in relation to the psychological support provided to them by hospice staff and its effectiveness in addressing their needs. Research is also needed to investigate the value of using a formal assessment tool over clinical judgement alone. Research on the effectiveness and implementation of training programmes for hospice staff is also recommended. Brief training programmes that can be delivered in a flexible format and include an ongoing supervision component may be of particular value. Lastly, hospice-employed staff are accustomed to caring for people approaching end of life on a daily basis and are confident and competent in initiating difficult conversations with patients and families; health care professionals working in hospitals often encounter people approaching end of life, [40] but are unlikely to have the same experience and confidence in providing psychological support to people with an advanced illness. Exploring the perspectives and training needs of hospital-based healthcare professionals would be useful.

## Strengths & Limitations

A mixed-methods design enabled a comprehensive understanding of staff perceptions, combining qualitative insights with quantitative evidence. The involvement of stakeholders in the refinement of the online questionnaire enhanced content validity by confirming the relevance and suitability of questions to the needs and concerns of hospice staff in providing psychological support, and ensuring no critical information was overlooked. The questionnaire was anonymous, minimising bias due to socially desirable responding. Unlike previous national surveys, the data was collected in May and June 2023, potentially representing a more current picture of how psychological support is provided compared with studies which involved data collection before [12] or during the onset of the COVID pandemic [14].

However, limitations should also be highlighted. The response rate was not possible to determine as we could not know how many eligible respondents had sight of the study invitation. Despite this, the overall sample size was higher than that reported in recent studies examining the psychological care in hospice settings, with other studies reporting 116[12] and 140[14] respondents. The study relied on convenience sampling. It is possible that hospice staff who are more inclined to provide psychological care, and perhaps more competent in it, may have been more likely to respond to the questionnaire than those less involved in psychological care. Furthermore, to preserve anonymity we did not ask respondents to state which hospice they worked in, which means that we do not know the total number of respondents by hospice. Despite clarifying eligibility, as this was advertised on social media, we cannot tell if any ineligible participants took part. The exclusion of palliative care staff based in hospital settings limits generalizability as the working environment, policies, training structures, patient demographics and goals of care differ markedly from hospices. It is possible that staff caring for terminally ill patients outside of hospice settings have greater training needs in relation to psychological support provision – this would be useful to explore in a future study. The competence and confidence scales were created for the purpose of the

present study; while the questionnaire had face validity, it had not been psychometrically tested for validity and reliability. Testing and further refinement of these scales would be a valuable future direction for research.

## Conclusion

Most hospice staff perceive themselves as competent and confident in providing basic psychological support. However, further training in the assessment and the delivery of psychological support is desired and would further enhance practice, especially in relation to caring for people with greater psychological support needs and navigating complex issues such as illness denial. Clearer guidance on referral for specialist psychological support is warranted.

## Supporting information

**S1 File. Supplementary File 1.**
(DOCX)

**S2 File. Supplementary File 2.**
(DOCX)

**S3 File. Supplementary File 3.**
(XLSX)

## Acknowledgments

Thank you to our stakeholders who played a key role in reviewing the questionnaire and making valuable suggestions.

## Author contributions

**Conceptualization:** Anne Finucane.

**Data curation:** Fatema Zavery, Sulafa Fakhreddin, Sarah Huibregtse Van Loon, Mhairi McDougall, Anne Finucane.

**Formal analysis:** Fatema Zavery, Sulafa Fakhreddin, Sarah Huibregtse Van Loon, Mhairi McDougall.

**Investigation:** Fatema Zavery, Sulafa Fakhreddin, Sarah Huibregtse Van Loon, Mhairi McDougall, Anne Finucane.

**Methodology:** Anne Finucane.

**Project administration:** Fatema Zavery, Sulafa Fakhreddin, Sarah Huibregtse Van Loon.

**Resources:** Anne Finucane.

**Supervision:** Anne Finucane.

**Visualization:** Fatema Zavery, Sulafa Fakhreddin, Sarah Huibregtse Van Loon, Anne Finucane.

**Writing – original draft:** Fatema Zavery, Sulafa Fakhreddin.

**Writing – review & editing:** Fatema Zavery, Sulafa Fakhreddin, Sarah Huibregtse Van Loon, Mhairi McDougall, Anne Finucane.

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
