## [Decision Letter · Decision Letter 0]

11 Apr 2025

Dear Dr. Finucane,

Thank you for submitting your manuscript to PLOS ONE. After careful consideration, we feel that it has merit but does not fully meet PLOS ONE’s publication criteria as it currently stands. Therefore, we invite you to submit a revised version of the manuscript that addresses the points raised during the review process.

We look forward to receiving your revised manuscript.

Kind regards,

Stefaan Six, Ph.D.

Academic Editor

PLOS ONE

2. Please ensure that you have specified a) Did participants provide their written or verbal informed consent to participate in this study?

b) If consent was verbal, please explain i) why written consent was not obtained, ii) how you documented participant consent, and iii) whether the ethics committees/IRB approved this consent procedure."

- In consent please state in Ethics Method section and manuscript if it is written or verbal. If consent was verbal, please explain a) why written consent was not obtained, b) how you documented participant consent, and c) whether the ethics committees/IRB approved this consent procedure.

Reviewers' comments:

Reviewer's Responses to Questions

**Comments to the Author**

1. Is the manuscript technically sound, and do the data support the conclusions?

Reviewer #1: Partly

Reviewer #2: Yes

Reviewer #3: Yes

2. Has the statistical analysis been performed appropriately and rigorously?

Reviewer #1: I Don't Know

Reviewer #2: Yes

Reviewer #3: Yes

3. Have the authors made all data underlying the findings in their manuscript fully available?

Reviewer #1: No

Reviewer #2: Yes

Reviewer #3: Yes

4. Is the manuscript presented in an intelligible fashion and written in standard English?

Reviewer #1: No

Reviewer #2: Yes

Reviewer #3: Yes

Reviewer #1: Dear Authors,

Congratulations on your great work! You have studied a critical area in health. There are some areas you should work on to improve the report.

Please rephrase the topic. Writing questionnaire in the topic seems quite inappropriate; Include population intervention (if applicable), context and outcome in the topic. According to your study, it seems that you have studied perceived competence and confidence.

Abstract: Specify study design, Rephrase eligible population, can use the term included participants, support results with figures

Introduction:

Cite who gave the projected data.

Link the information provided in the introduction, make it more oragnised.

Method: was this a parallel mixed method? Please specify

Participants: who were your target population?what was their total number in each of the hospices? You mentioned that you excluded some members(like mental health professionals, and

staff affiliated with NHS-based hospices) because you did not get the ethical approval for it. Did you wanted to collect data from these professionasls but IRB/C did not allow you to? What might be its adverse effect on the validity of the study?

Include the information about total number of participants in the study here rather than in the results

You stated that you excluded the reponse with more than 50% missing data? how many questionnaire did you get with missing data less than 50%?, How did you manage it? How did you manage your online data?

What was the trend of your data? why did you choose t-test and ANOVA?

Describe in detail about qualitative data analysis. How did you prevent bias in data analysis of qualitative studies?

Results:

N= Population, n=sample; replace N with n in table heading and table

Are the tables formatted and hyperlinked according to guidelines?

No need to write number while describing tables, you can use only percentage but be clear where you keep it.

What does "other" mean? was this term stated by participants??

Tables need formatting

General:

Correct typo error and grammatical inconsistencies.

All the best!

Reviewer #2: I sincerely appreciate the opportunity to review the manuscript entitled “Providing psychological support to people impacted by terminal illness: A questionnaire study of hospice staff confidence and training needs.”

I would like to begin by highlighting the relevance and pertinence of the work presented. The study addresses a fundamental and often underestimated aspect of palliative care: the preparedness of hospice staff to offer psychological support to patients and families in contexts of terminal illness.

Below, I provide some specific comments that I believe can contribute to improving the clarity, depth, and impact of the manuscript.

In the title it does not make sense to use the term questionnaire, since it does not indicate the research methodology. Therefore, I think it would be better to replace it by descriptive mixed methods study.

Missing information on how to access the study data, please add the correct information when uploading the data to the platform instead of XXX.

In the introduction, the sentence “In Australia, guidelines recommend a 0.15 full-time...” is not understood. Please explain what the 0.15 refers to?

Explain abbreviations and acronyms throughout the text, such as NICE, NHS, JISC, EOLC....

Expand the introduction with the importance of training healthcare personnel in psychology skills and knowledge, especially in palliative care.

The methodology lacks the calculation of the sample size, based on other studies we can know if we arrive at a sample calculation that is representative of the population under study. There are tools such as the GRANMO calculator that help to perform this type of calculation.

It is not clear whether or not any software was used in the analysis of qualitative data, nor if data saturation was reached.

In the results there are typos such as “ove to r” and in the respondent's response 21 occupational therapist.

The tables should be adjusted to the page size to improve the readability and comprehension of the data.

As well as the quality of the figures, since they look blurry and this hinders reading.

Reviewer #3: The document lacks defined acronyms which need to be defined when first presented for readability and explanation to readers who have not seen or are aware of the acronym. The document interchanges palliative care and hospice care and they are not the same thing. More attention within the explanation needs to address that hospice clinicians provide palliative care to their patients and caregivers, but palliative care clinicians are not specifically hospice clinicians. Palliative care clinicians can take care of serious-illness patients that are not at the end of life or enrolled in hospice. There are some typographical errors and need to be corrected.

**Do you want your identity to be public for this peer review?** For information about this choice, including consent withdrawal, please see our Privacy Policy

Reviewer #1: **Yes: ** Mira Adhikari Baral

Reviewer #2: **Yes: ** Esperanza Barroso-Corroto

Reviewer #3: No

---

## [Author Response · Author response to Decision Letter 1]

26 May 2025

We have attached a response to Reviewers file detailing how we have responded to each comment and suggestion.

---

## [Decision Letter · Decision Letter 1]

23 Jun 2025

Dear Dr. Finucane,

We look forward to receiving your revised manuscript.

Kind regards,

Stefaan Six, Ph.D.

Academic Editor

PLOS ONE

Journal Requirements:

Reviewers' comments:

Reviewer's Responses to Questions

**Comments to the Author**

Reviewer #1: All comments have been addressed

Reviewer #2: All comments have been addressed

Reviewer #3: All comments have been addressed

2. Is the manuscript technically sound, and do the data support the conclusions?

Reviewer #1: Partly

Reviewer #2: Yes

Reviewer #3: Yes

3. Has the statistical analysis been performed appropriately and rigorously?

Reviewer #1: I Don't Know

Reviewer #2: Yes

Reviewer #3: Yes

4. Have the authors made all data underlying the findings in their manuscript fully available?

Reviewer #1: Yes

Reviewer #2: Yes

Reviewer #3: Yes

5. Is the manuscript presented in an intelligible fashion and written in standard English?

Reviewer #1: Yes

Reviewer #2: Yes

Reviewer #3: Yes

Reviewer #1: Overall: Congratulations! The manuscript looks better now

Abstract:

Setting /participants: free-text data was analysed thematically guided by which framework approach. Please specify

Methods

Sample size: write about the sampe size first and then only describe how you derived it. OR did you approached for complete enumeration?

Questionnaire deelopment: why did you include demographic information at the last part of the questionnaire? c

You mentioned that the tool has only face alidity. Did you check for content validity of the tool?

Participant recruitment: how did you ensure theat the non-elligible participants did not fill in the questionnaire? In the social media

Table 1: what does N mean with regards to the table?

Figure 2: had statistically higher perceived competence scores than those who did not (M = 21.73, SD = 3.57), 95% CI [3.47, 6.18] (t (145) = 7.04, p <0.01)= is it necessay to include all the data in your result?

Rewrite table according to the PLOS one guidelines. Have a look at other published articles. For example use small n instead for N as N means population, change format of tables

Table three: no need for another column, you can write n at each items of training received

Discussion: well written

Reviewer #2: I would like to extend my sincere congratulations on the work presented in your manuscript. The study reflects a commendable research effort, characterized by methodological rigor and a clear contribution to the body of knowledge in your field.

Reviewer #3: (No Response)

**Do you want your identity to be public for this peer review?** For information about this choice, including consent withdrawal, please see our Privacy Policy

Reviewer #1: No

Reviewer #2: **Yes: ** Esperanza Barroso-Corroto

Reviewer #3: **Yes: ** Patricia Natalia Brothers

---

## [Author Response · Author response to Decision Letter 2]

14 Jul 2025

Dear Editor,

Thank you very much for considering our manuscript. We have considered each reviewer comment in turn and have amended our manuscript to increase clarity in response to these comments. We are very grateful to the reviewers for their thoughtful suggestions which have improved our manuscript. We also removed one reference which was no longer relevant, so the bibliography has been re-ordered very slightly.

Our point-by-point response to reviewers is outlined in the 'response to reviewers' file that we attach with this submission.

Yours sincerely

Anne Finucane and Fatema Zavery

---

## [Decision Letter · Decision Letter 2]

20 Aug 2025

Providing psychological support to people impacted by terminal illness: A  mixed methods study of hospice staff perceived competence, confidence and training needs

PONE-D-25-05121R2

Dear Dr. Finucane,

We’re pleased to inform you that your manuscript has been judged scientifically suitable for publication and will be formally accepted for publication once it meets all outstanding technical requirements.

Kind regards,

Antony Bayer

Academic Editor

PLOS ONE

Additional Editor Comments (optional):

Reviewers' comments:

Reviewer's Responses to Questions

**Comments to the Author**

Reviewer #1: (No Response)

Reviewer #2: All comments have been addressed

Reviewer #3: All comments have been addressed

2. Is the manuscript technically sound, and do the data support the conclusions?

Reviewer #1: Yes

Reviewer #2: Yes

Reviewer #3: Yes

3. Has the statistical analysis been performed appropriately and rigorously?

Reviewer #1: I Don't Know

Reviewer #2: Yes

Reviewer #3: Yes

4. Have the authors made all data underlying the findings in their manuscript fully available?

Reviewer #1: Yes

Reviewer #2: Yes

Reviewer #3: Yes

5. Is the manuscript presented in an intelligible fashion and written in standard English?

Reviewer #1: Yes

Reviewer #2: Yes

Reviewer #3: Yes

Reviewer #1: (No Response)

Reviewer #2: Congratulation about your work. The coments has been correctly adressed and the manuscript has been improved.

Reviewer #3: All comments have been addressed. Thank you to the authors for addressing and sharing this important work.

**Do you want your identity to be public for this peer review?** For information about this choice, including consent withdrawal, please see our Privacy Policy

Reviewer #1: **Yes: ** Mira Adhikari Baral

Reviewer #2: **Yes: ** Esperanza Barroso-Corroto

Reviewer #3: No

---

## [Editor Report · Acceptance letter]

PONE-D-25-05121R2

PLOS ONE

Dear Dr. Finucane,

I'm pleased to inform you that your manuscript has been deemed suitable for publication in PLOS ONE. Congratulations! Your manuscript is now being handed over to our production team.

Kind regards,

on behalf of

Professor Antony Bayer

Academic Editor

PLOS ONE